# Role of the Mitochondrial Citrate-malate Shuttle in *Hras12V*-Induced Hepatocarcinogenesis: A Metabolomics-Based Analysis

**DOI:** 10.3390/metabo10050193

**Published:** 2020-05-13

**Authors:** Chuanyi Lei, Jun Chen, Huiling Li, Tingting Fan, Xu Zheng, Hong Wang, Nan Zhang, Yang Liu, Xiaoqin Luo, Jingyu Wang, Aiguo Wang

**Affiliations:** Laboratory Animal Center, Department of Comparative Medicine, Dalian Medical University, Dalian 116044, Liaoning, China; L727947980@163.com (C.L.); chenjun@dmu.edu.cn (J.C.); lhl@dmu.edu.cn (H.L.); fantt2009@126.com (T.F.); zhengxu221@163.com (X.Z.); 18202990141@163.com (H.W.); 18018929690@163.com (N.Z.); 18834072645@163.com (Y.L.); 18434371872@163.com (X.L.)

**Keywords:** hepatocarcinogenesis, *Ras* oncogene, metabolomics, transcriptomics, mitochondrial citrate-malate shuttle

## Abstract

The activation of the Ras signaling pathway is a crucial process in hepatocarcinogenesis. Till now, no reports have scrutinized the role of dynamic metabolic changes in *Ras* oncogene-induced transition of the normal and precancerous liver cells to hepatocellular carcinoma in vivo. In the current study, we attempted a comprehensive investigation of *Hras12V* transgenic mice (*Ras*-Tg) by concatenating nontargeted metabolomics, transcriptomics analysis, and targeted-metabolomics incorporating [U-^13^C] glucose. A total of 631 peaks were detected, out of which 555 metabolites were screened. Besides, a total of 122 differently expressed metabolites (DEMs) were identified, and they were categorized and subtyped with the help of variation tendency analysis of the normal (W), precancerous (P), and hepatocellular carcinoma (T) liver tissues. Thus, the positive or negative association between metabolites and the hepatocellular carcinoma and *Ras* oncogene were identified. The bioinformatics analysis elucidated the hepatocarcinogenesis-associated significant metabolic pathways: glycolysis, mitochondrial citrate-malate shuttle, lipid biosynthesis, pentose phosphate pathway (PPP), cholesterol and bile acid biosynthesis, and glutathione metabolism. The key metabolites and enzymes identified in this analysis were further validated. Moreover, we confirmed the PPP, glycolysis, and conversion of pyruvate to cytosol acetyl-CoA by mitochondrial citrate-malate shuttle, in vivo, by incorporating [U-^13^C] glucose. In summary, the current study presented the comprehensive bioinformatics analysis, depicting the *Ras* oncogene-induced dynamic metabolite variations in hepatocarcinogenesis. A significant finding of our study was that the mitochondrial citrate-malate shuttle plays a crucial role in detoxification of lactic acid, maintenance of mitochondrial integrity, and enhancement of lipid biosynthesis, which, in turn, promotes hepatocarcinogenesis.

## 1. Introduction

Metabolism is a fundamental biological process in the normal as well as cancerous cells, and metabolic alterations are perceived as a hallmark of cancer [1]. However, the complex biochemical pathways regulation, as well as cellular and molecular heterogeneities within and across tumor entities, impede the elucidation of altered metabolism in cancer cells [1]. Recently, besides the studies primarily targeting the role of the Warburg effect in cancer, many valuable evidences on TCA cycle rewiring, glutamine metabolism, glutaminolysis, serine biosynthesis/one-carbon pathway, 2-hydroxyglutarate production, etc. have revealed new principles of cancerous metabolism and shed light on carcinogenesis. Thus, deepening molecular mechanisms underlying the tumor’s metabolic characteristics will lead to improved tumor categorization and identification of the potential therapeutic agent in cancer [2].

Liver cancer is a global health concern by virtue of its rising incidence and low survival rate [3,4]. Hepatocellular carcinoma (HCC) accounts for over 80% of liver cancer cases. Apart from being highly malignant, recurrent, and drug-resistant, it is often diagnosed at an advanced stage [5]. For these reasons, the need to identify molecular features that uniquely define or contribute to HCC progression remains clinically urgent. As the liver functions as a major digestive gland and is the center of systemic metabolism in the body, liver cancer transformation is coupled with prominent metabolic alterations. Thus, identification of the metabolites that explicitly define or promote HCC progression demands immediate clinical attention.

Nontargeted metabolomics is the method of choice for the investigation of the carcinogenesis mechanism and the identification of novel biomarkers comprising HCC [6,7]. Targeted metabolomics or stable isotope resolved metabolomics (SIRM) evaluates an isotope-filtered selection of molecules and leads to an improved understanding of the dynamics and compartmentation of metabolic pathways and networks [8]. The isotope-labeled tracer [U-^13^C]glucose is majorly employed for labeling the metabolic intermediates of tricarboxylic acid (TCA) cycle and amino acids, as well as for comprehension of the metabolic pathways, such as glycolytic and pentose phosphate pathway [9]. Tissue metabolomics enacts pairwise comparison of precancerous and cancerous tissues from each organism to remove individual variation. Recent advances have been witnessed in the field of cancer metabolism with the application of unbiased and targeted metabolomics along with the genetic and biochemical studies using animal models.

In murine HCC, *Ras* mutation was reported in 70% of chemically-induced and spontaneous cases [10]. Although the occurrence of mutational activation of the Ras protein is relatively low (~5%) in human HCC, the receptor-mediated hyperactivation of the RAS-dependent signal transduction pathway is a frequent event [11,12,13]. These pieces of evidence validate the crucial role of RAS in hepatocarcinogenesis [14]. We generated the *Hras12V* transgenic mouse lineage (*Ras*-Tg) with the hepatocyte-specific expression of *RAS* oncogene, which resulted in multicentric spontaneous hepatic tumorigenesis with a high level of reproducibility [14]. This liver tumor model has helped other researchers and us to unravel significant findings [15,16]. Therefore, the *Ras*-Tg is a relevant model for the investigation of the underlying mechanisms in hepatocarcinogenesis.

In the current study, we demonstrated the global metabolic alterations related to *Ras* oncogene-induced hepatic tumorigenesis with the help of the *Ras*-Tg and nontargeted metabolomics and transcriptomics analysis. Furthermore, the [U-^13^C] glucose targeted metabolomics confirmed the flux of glucose in the glycolysis, the pentose phosphate pathway, and the TCA cycle. Interestingly, the mitochondrial citrate-malate shuttle was identified as a vital link between the enhanced lipid biosynthesis and glycolysis, where glycolysis eliminated lactic acid production.

## 2. Materials and Methods

### 2.1. Animals, Sampling, and Histopathological Examination

We used *Hras12V* transgenic mice (*Ras*-Tg) in this study to investigate the *Ras* oncogene-induced metabolic changes in hepatocarcinogenesis. Nine-month-old *Ras*-Tg males and C57BL/6J wild-type non-transgenic males (Non-Tg) (*n* = 8 in each group) were sacrificed. We harvested the normal (W) of non-Tg, precancerous (P), and hepatocellular carcinoma (T) of *Ras*-Tg liver samples. A portion of the harvested liver sample was immediately flash-frozen in liquid nitrogen. The histopathological investigation was performed with the remaining liver sample based on the criteria described by Frith and Ward [17], and the detailed information related to the histopathological alterations are presented in the Appendix A. Only the pathologically confirmed liver samples were used for subsequent experimental procedures (Appendix A). The current study was ethically approved by the Institutional Animal Care and Use Committee of Dalian Medical University.

### 2.2. Experimental Design

We performed the metabolomic investigation of the harvested liver samples to unravel the metabolic pathways associated with hepatocarcinogenesis. The altered metabolites among W, P, and T were identified in the metabolomic analysis. Additionally, the positively and negatively associated metabolites of HCC and *Ras* oncogene were identified by an innovative expression-change-pattern analysis. We integrated these findings with the mRNA sequencing data. By the comprehensive analysis of bioinformatics data and validation of the variation in the key metabolites and genes levels, the significantly affected metabolic pathways were elucidated in hepatocarcinogenesis. Among these pathways, the dynamic glucose metabolism fate was further confirmed by metabolomic analysis with labeled [U-^13^C] glucose (Appendix A).

Please see Appendix A for more detailed information of this section.

## 3. Results

### 3.1. Differential Metabolites and Pathways Identified by Metabolomic Analysis in Hepatocarcinogenesis

The typical GC-MS total ion current (TIC) chromatograms of extracts from W, P, and T were analyzed before clustering the samples. The data showed that the retention time of the internal standard was consistent in each run, and distinct chromatogram patterns of W, P, and T were observed (Appendix A). We screened a total of 555 metabolites from the 631 peaks identified by the interquartile range denoising method. 

The principal component analysis (PCA) calculations were represented as the SIMCA 3D plot and two-dimensional plan. The close clustered dots depicted similar composition of the samples in the same groups, different composition of the samples in different groups, and the different profiles of the W, P, and T in different groups (Appendix A). The robustness of the models was shown by the partial least squares-discriminant analysis (PLS-DA) score plots, which validated the calculation analysis of the model with a low risk of overfitting and reliability (Appendix A). We identified significantly changed metabolites (Appendix A)—56 metabolites in the W/P, 69 metabolites in the P/T, and 90 metabolites in the W/T—in the orthogonal projections to latent structures-discriminant analysis (OPLS-DA) analysis with the criterion of *p* < 0.05 (Appendix A). Furthermore, we performed metabolite set enrichment analysis (MSEA) to identify the critical metabolic pathways associated with hepatocarcinogenesis. It led to the enrichment of 37 pathways (W/P), 43 pathways (P/T), 47 pathways (W/T) related to the metabolisms of galactose, citrate, starch, sucrose, alanine, aspartate, and glutamate, and so on (Appendix A). 

### 3.2. Variation Tendencies of Metabolites Identified in Hepatocarcinogenesis

We categorized the altered metabolites into four groups, i.e., positively or negatively associated metabolites of HCC and RAS, to demonstrate their variation tendencies in *Ras* oncogene-induced hepatic tumorigenesis (Figure 1). 

The positively associated metabolites in HCC were significantly up-regulated in T in contrast to P or W. The metabolites enlisted in this group could be categorized into principally three types. (i) T > P = W, most of the metabolites belonged to this category, which indicated that the metabolites were positively associated with HCC; (ii) T > W > P and (iii) W > T > P, which indicated their precedence in the hepatic tumorigenesis, and so they were suppressed by the cancer-defense system in precancerous hepatocytes (Figure 1A). Among these metabolites, glutamic acid and linoleic acid were reported to be positively associated with the HCC [18,19].

The metabolites negatively associated with the HCC were significantly down-regulated in T as compared to W or P. The metabolites enlisted in this category were categorized into three types: (i) T < P = W, which indicated their absolute HCC negative association; (ii) T < W < P and (iii) W < T < P, which indicated their importance in the anti-hepatic tumorigenesis, and hence they were overexpressed by the cancer-defense system in precancerous hepatocytes (Figure 1B). Among these metabolites, taurine was reported to be negatively associated with HCC [20].

The metabolites positively associated with *Ras* oncogene were significantly up-regulated in P and T than W, and so these metabolites responded positively to the expression of *Ras* oncogene. The metabolites enlisted in this category could be categorized into two types. (i) T > P > W and (ii) T = P > W (Figure 1C). 

The metabolites negatively associated with *Ras* oncogene were significantly down-regulated in P and T than W, and so these metabolites responded negatively to the expression of *Ras* oncogene. The metabolites enlisted in this category could be categorized into two types: (i) T < P < W and (ii) T = P < W (Figure 1D). 

### 3.3. Differential Genes and Pathways Identification by Transcriptomics Analysis in Hepatocarcinogenesis

We investigated the mRNA expression profiles in hepatocarcinogenesis with the help of next-generation sequencing (NGS) and analyzed the differential expression of genes (DEGs) by the pairwise comparison between W, P, and T. This analysis led to the identification of a total of 21,123 genes, of which 3839 were identified as DEGs with a cut-off ratio ≥ 1.5 or ≤ 0.67 and *p* < 0.05. We identified 691 DEGs from a comparison between P and W, of which 306 were up-regulated, and 385 were down-regulated in P as compared to W. Similarly, a fold-change study between T and W led to the identification of 1690 DEGs, of which 1018 were up-regulated, and 635 were down-regulated in T when compared to W. In case of a P and T, 1458 DEGs were identified, of which 912 were up-regulated, and 546 were down-regulated in P as compared to T (Appendix A). Besides, we also performed cluster enrichment analysis with the help of the Kyoto Encyclopedia of Genes and Genomes (KEGG) pathways on DAVID, a bioinformatics platform (http://david.abcc.ncifcrf.gov/), to analyze the biological significance of the DEGs. The outcome of detailed bioinformatics analysis is depicted in Appendix A. The KEGG pathway enrichment analysis of the W/P DEGs revealed the up-regulation of four pathways (drug metabolism, biosynthesis of unsaturated fatty acids, PPAR signaling pathway, retinol metabolism) and down-regulation of eight pathways (PPAR signaling pathway, fatty acid metabolism, glutathione metabolism, and so on). The P/T DEGs pathway analysis revealed the up-regulation of 18 pathways (alanine, aspartate and glutamate metabolism, MAPK signaling pathway, fatty acid metabolism, and so on) and down-regulation of 26 pathways (PPAR signaling pathway, pyruvate metabolism, glycolysis/gluconeogenesis, and so on). In the W/T, DEGs pathway analysis demonstrated up-regulation of 19 pathways (PPAR signaling pathway, pathways in cancer, p53 signaling pathway, and so on) and down-regulation of 32 pathways (PPAR signaling pathway, pyruvate metabolism, glycolysis/gluconeogenesis, and so on). 

### 3.4. Mitochondrial Citrate Malate Shuttle and Lipid Biosynthesis Enhanced during Hepatocarcinogenesis

We integrated the metabolomics and transcriptomics data to excavate the *Ras* oncogene-induced key metabolic pathways in hepatocarcinogenesis. We selected the genes and metabolites from the identified key pathway, which were significantly altered at least twice in the pairwise comparison between W, P, and T. The key genes and metabolites were further validated by RT-qPCR, Western blot, and corresponding detection kits.

#### 3.4.1. Glycolysis 

The key metabolites in glycolysis, such as glucose-1P, glucose-6P, fructose-6P, and fructose, were significantly elevated in P as compared to W. The altered level of these metabolites might be the result of decreased *G6pc*, which mitigates the glucose-1P conversion to glucose. However, *Pkm2*, pyruvate, and lactic acid levels were not changed between W and P. It indicated that a portion of elevated fructose-1,6P2 took part in other metabolic pathways before getting converted to pyruvate. In T, the crucial glycolytic metabolites, such as glucose-1P, glucose-6P, and fructose-6P, were significantly elevated as compared to P, which might be the outcome of both down-regulated *G6pc* and up-regulated *Pkm2* as compared to P. This resulted in a significant accumulation of pyruvate. Surprisingly, reduced lactate level along with the absence of monocarboxylate transporter 1/4 (MCT1/4) and CD147 overexpression in T pointed towards the presence of a novel mechanism in hepatoma, which promoted the mitochondrial pyruvate entry and prevented the lactic acid-induced cytotoxicity (Figure 2A, Figure 3A, Figure 4A, respectively; Appendix A; Table 1 and Table 2; Appendix A.

#### 3.4.2. TCA Cycle and Citrate Malate Shuttle

In P, the decreased level of fumarate and succinate indicated the blocked TCA cycle as compared to W. The pyruvate, citrate, and malate levels were not altered between W and P, which indicated that TCA cycle-related citrate malate shuttle was not significantly affected. In T, the decreased level of succinate as compared to W indicated the blocked TCA cycle. Intriguingly, the notably escalated levels of pyruvate, citrate, and malate demonstrated the enhanced TCA cycle-related citrate malate shuttle in T than W and P (Figure 2A, Figure 3A, Figure 4A, respectively; Appendix A; Table 1 and Table 2; Appendix A). Additionally, as the mitochondrial carrier SLC25A1 is the specific mitochondrial transporter of citrate/malate, we further verified its mRNA levels in W, P, and T. Consistent to the next-generation sequencing (NGS) data, there was no significant change for the mRNA levels of SLC25A1 among W, P, and T (data not shown). It might indicate that the expression level of SLC25A1 was sufficient to support citrate/malate transport and had not been specially regulated in the present experimental system.

#### 3.4.3. Lipid Biosynthesis 

In P, we noticed a significant up-regulation of *Acly,* but *Lss* and *Acsl4* expression levels were unaltered. Besides, the down-regulation of *Dhcr7* and *Mvd* coupled with unaltered TG content as compared to W indicated significantly disturbed lipid biosynthesis in P. However, in T, the elevated level of TG and enhanced expression of lipid synthesis enzymes genes—*Lss, Acsl4, Dhcr7, Mvd*—indicated increased lipid biosynthesis as compared to W and P (Figure 2A, Figure 3A, Figure 4A, respectively; Appendix A; Table 1 and Table 2; Appendix A).

### 3.5. Alteration of Pentose Phosphate Pathway (PPP), Cholesterol and Bile Acid Metabolism, Glutathione Metabolism during Hepatocarcinogenesis

#### 3.5.1. PPP

Hydride anion is a crucial component in the synthesis of biomolecules and primarily provided by reduced form of nicotinamide adenine dinucleotide phosphate (NADPH) from PPP. In P, we did not observe any variation in the expression level of *Pgd*, *G6pd*, *Pkm2,* and NADPH level between W and P, which indicated the absence of *Ras* oncogene effect. However, in case of T, we found significantly overexpressed *Pgd*, *G6pd*, *Pkm2,* and NADPH and significantly decreased oxidation form of reduced nicotinamide adenine dinucleotide phosphate (NADP^+^), and Xanthosine levels as compared to W and P. It indicated that escalated level of PPP in T contributed towards the synthesis of nucleic acid, lipid, cholesterol, and so on (Figure 2A, Figure 3A, Figure 4A, respectively; Appendix A; Table 1 and Table 2; Appendix A).

#### 3.5.2. Cholesterol and Bile Acid Metabolism

We observed a significant down-regulation of *Cyp27a1*, *Cyp7b1*, *cyp8b1*, *Hsd3b7*, and *Baat* in P along with the decreased level of cholate and glycine. It resulted in the cholesterol accumulation as compared to the W. The expression levels of *HMGCR*, *Lss*, *Dhcr7*, and *Mvd,* which coded for cholesterol biosynthetic enzymes, were either unaltered or decreased. It indicated the insignificant contribution of the cholesterol synthesis in the cholesterol accumulation in P. In case of T, we observed the overexpression of genes, which coded for cholesterol biosynthetic enzymes, i.e., *Hmgcr*, *Lss*, *Dhcr7*, *Mvd*, and the down-regulation of genes, which coded for bile acid synthesis enzymes, i.e., *Cyp8b1*, *Hsd3b7*, *Cyp27a1*, *Cyp7b1*, *Baat*, as compared to P or W. Additionally, due to the insufficient supply of glycine and taurine, the content of cholate was diminished in T as compared to P. Besides, the content of cholesterol was significantly higher in T than W but lower than P. It indicated that hepatoma cells had alternate mechanisms for cholesterol level reduction (Figure 2B, Figure 3B, Figure 4B, respectively; Appendix A; Table 1 and Table 2; Appendix A).

#### 3.5.3. Glutathione Metabolism

In P, we did not observe any change in the level of L-glutamate and *Gss* expression as compared to W, which explained the unaltered level of glutathione (GSH) in P. However, decreased NADP^+^ level and unaltered NADPH and expression levels of *Gss*, *Pgd*, *G6pd*, *Gpx4* resulted in the increased ROS (reactive oxygen species) level in P in the presence of *Ras* oncogene as compared to W. In T, the escalated level of 5-oxaproline and L-glutamate along with overexpressed GSS led to increased GSH level in T as compared to P and W. Additionally, the down-regulated *Abat* impaired the succinate flow. Besides, the increased *Anpep* expression resulted in a sufficient supply of cys-Gly that also contributed to the higher level of GSH. The diminished NADP^+^ level and elevated NADPH level along with overexpressed *Gss*, *Pgd*, *G6pd*, and *Gpx4* led to the decreased ROS level in T than P when exposed to the higher levels of *Ras* oncogene. It indicated that hepatoma recruited the GSH system to neutralize *Ras* oncogene-induced ROS level to promote tumor cell growth. However, precancerous hepatocytes with *Ras* oncogene did not follow the same mechanism and so demonstrated a higher level of ROS, which, in turn, induced cellular stress in cells and contributed to tumorigenesis (Figure 2C, Figure 3C, Figure 4C, respectively; Appendix A; Table 1 and Table 2; Appendix A).

### 3.6. Enhanced Mitochondrial Citrate-Malate Shuttle and PPP in Hepatocarcinogenesis Confirmed by In Vivo Assay of [U-^13^C] Glucose Metabolism 

We performed the metabolic flux analysis by examining metabolic fates of the [U-^13^C]-labeled glucose to validate the in vivo alteration of metabolic pathways in HCC in *Ras*-Tg mice as compared to the wild type mice. The outcome of the analysis showed that the glycolysis, TCA cycle, mitochondrial citrate-malate shuttle, and PPP were significantly altered in hepatocarcinogenesis (Figure 5; Table 3; Appendix A).

#### 3.6.1. Glycolysis

We observed that the level of [U-^13^C]-labeled glucose and other intermediates of the glycolytic pathway, such as glucose 6P, fructose 6P, 3-phosphoglycerate, phosphoenolpyruvate, and pyruvate, were significantly reduced in P and T as compared to W. It indicated the significant enhancement of the glycolytic pathway in the liver of *Ras*-Tg, specifically in T. Intriguingly, the [U-^13^C]-labeled lactate was found to be significantly reduced in T but not in P. It indicated that the HCC enacted alternative mechanism to mitigate the lactate toxicity by enhancing glycolysis. In line with this finding, the citrate content was found to be significantly escalated in T, which indicated that the pyruvate was readily converted to citrate by TCA cycle in hepatoma. 

#### 3.6.2. TCA Cycle and Citrate-malate Shuttle

We observed that the citrate content was not altered in P as compared to W; however, the succinate, fumarate, and malate were significantly reduced as compared to W. It indicated that the TCA was significantly inhibited in normal hepatocytes exposed to *Ras* oncogene. In case of T, the TCA cycle citrate content was significantly increased as compared to P and W, and the succinate content was significantly decreased as compared to W. It indicated the blocked TCA cycle in T. The explicit and significant rise in citrate, malate, and fumarate level indicated enhanced citrate-malate shuttle and perpetual conversion of cytoplasmic pyruvate into acetyl-CoA to suppress the cytotoxic pyruvate to lactate conversion. The cytoplasmic acetyl-CoA was further utilized in lipid synthesis and energy generation. 

#### 3.6.3. PPP

We observed the content of [U-^13^C]-labeled ribose-5-phosphate, sedoheptulose-7-phosphate, and erythrose-4-phosphate was significantly reduced in P as compared to W. It indicated the blocked PPP in normal hepatocytes when exposed to *Ras* oncogene. However, in the case of T, the [U-^13^C]-labeled ribose-5-phosphate level was significantly escalated, and its downstream metabolites’—sedoheptulose-7-phosphate and erythrose-4-phosphate—levels were significantly diminished as compared to P. It indicated that PPP was regulated to provide sufficient ribose-5-phosphate for nucleic acid synthesis and HCC proliferation. 

## 4. Discussion

Clinically, the complex pathogenesis, symptoms, and developmental stages in HCC patients restrict the matched specimen collection and the systemic investigation for the identification of hepatocarcinoma etiology. Therefore, animal models provide crucial information about the underlying mechanisms of hepatic tumorigenesis and its development induced by a specific pathogen. Especially, the matched wild-type and precancerous liver of hepatic tumor allow dynamic variation analysis in hepatocarcinogenesis. Similarly, *Ras*-Tg mice provide crucial in vivo conditions for an in-depth investigation of metabolic alteration in *Ras* oncogene-induced hepatocarcinogenesis. The *Ras* oncogene is expressed in all hepatocytes and leads to countable tumors development, which indicates that the strong defense system exists to prevent hepatic tumor development. Therefore, precancerous hepatocyte characteristics play a significant role in the elucidation of tumorigenesis. 

In the current study, we classified the DEMs expression patterns into multiple categories and subtypes and identified the positively or negatively associated metabolites of HCC and RAS oncogene, especially anti-tumor defense system-regulated metabolites (Figure 1). Among these DEMs, few metabolites were reported to have consistent variation trends in human HCC, which indicated the reliability of the expression pattern analysis, and the rest of the metabolites remained unexplored. These novel findings will provide valuable clues to the researches. For instance, glycine, nicotinamide, adenosine [21,22,23] have been reported to be negatively associated with HCC, whereas ribose-5-phosphate, arachidonic acid [24,25] are positively associated with HCC. Interestingly, in the current study, we also found the negatively and positively associated metabolites in the presence of *Ras* oncogene. The outcome of our study validated the importance of these metabolites in the hepatocarcinogenesis and elucidated their correlation with activated RAS signaling pathways.

The cancerous and normal proliferating cells employ glucose catabolism to provide precursor molecules and to reduce equivalents in the form of NADPH and meet the requirement of a modest increase in ATP consumption. The cancerous cell uses two different mechanisms to ensure efficient glucose catabolism and to impede the excessive pyruvate accumulation. The first mechanism is by aerobic glycolysis, also known as the Warburg effect. The TCA cycle generates NADH and ATP, the primary feedback, and the negative regulator of glucose metabolism. The cancer cells convert excessive pyruvate to lactate and protons rather than transporting them to the mitochondria for oxidative phosphorylation maintenance. Then, the excessive lactate and protons are transported to extracellular space by monocarboxylate transporters (MCTs) and MCT1/4 chaperone CD147. It prevents the acid-induced apoptosis and creates an extracellular acidic environment, which suppresses the effect of the immune system and favors tumor invasion through the activation of metalloproteinases [26].

The second mechanism involves the citrate-malate shuttle pathway, which converts excessive pyruvate to acetyl-CoA and occurs in mitochondria. Acetyl-CoA serves as a precursor in lipid biosynthesis and protein acetylation. In this pathway, pyruvate enters the tricarboxylic acid (TCA) cycle and gets converted to citrate. It is secreted into the cytosol through mitochondrial tricarboxylate carrier and broken down into acetyl-CoA and oxaloacetate where oxaloacetate is converted into malate and re-imported into mitochondria to maintain anaplerosis. The conversion of mitochondrial malate to oxaloacetate generates electrons that enter the mitochondrial electron transport chain through glycerol phosphate dehydrogenase activity and complex I. These electrons maintain mitochondrial integrity and ATP production despite reduced catabolic TCA cycle activity [2].

However, the functional outcome of these two pathways in cancer cells is influenced by the microenvironmental conditions, cancer types and subclasses, genetic alterations, or deregulations of tumor suppressors or oncogenes. In primary HCC, the percentage of MCT1 positive-expressed hepatoma cells decreases from 96.2% in the non-neoplastic cells to 63.0% in the primary HCC. However, the percentage of MCT4 positive-expression is 11.5% in the non-neoplastic to 40.8% in the primary HCC. Additionally, the positive expression percentage of corresponding chaperone CD147 also decreases from 84.6% in non-neoplastic to 48.0% in primary HCC [27]. Moreover, the MCT1/4 and CD147 also did not increase in T of *Ras*-Tg (Appendix A). Till now, no studies have reported the therapeutic role of MCT inhibitors in HCC therapy. It shows that the MCTs system is not efficiently explored as far as the efflux of excessive lactate and protons in the majority of primary HCC is concerned. Therefore, substantial HCC may primarily involve mitochondrial citrate-malate shuttle for excessive pyruvate elimination instead of aerobic glycolysis so as to prevent apoptosis due to inefficient efflux of excessive lactate and protons.

The elevated de novo lipid biosynthesis is one of the hallmarks of HCC and precancerous lesions irrespective of the serum lipid levels. Fatty acids serve as explicit and significant risk factors in hepatocarcinogenesis. Recent studies have shown that both de novo synthesized and exogenous fatty acids support the growth of HCCs [28]. Additionally, fat deposition is often seen in 36% of well-differentiated early-stage HCCs with the size range of 1.1–1.5 cm [29,30] and with an increased lipid droplet content [31]. In the present study, we found that the *Ras* oncogene-induced HCC demonstrated increased glycolysis and decreased lactate levels with massive lipid deposition. It indicated that de novo lipid biosynthesis was a crucial alternative pathway in HCC, which not only eliminated excessive pyruvate but also supported hepatocarcinogenesis.

The TCA cycle is a central route of oxidative phosphorylation in cells to meet their bioenergetic, biosynthetic, and redox balance requirements [32,33]. However, the role of the TCA cycle in cancer metabolism, tumorigenesis, and cancer cell proliferation was neglected due to the prevailing notion that carcinogenic cells primarily utilize aerobic glycolysis to adapt to the damaged mitochondrial respiration. In recent years, a large amount of evidence has highlighted the relevance of the TCA cycle rewiring in a variety of cancers, including HCC [33]. The TCA cycle rewiring by expression and activity dysregulation of enzymes plays a central role in cancer cells for the traffic of molecules between mitochondria and cytosol to supply building blocks for lipids, nucleic acids, and protein synthesis. Therefore, the aerobic glycolysis is not an adaptive condition but a tightly regulated metabolic state, supporting an increased biosynthetic demand [33]. Accordingly, the recent advances have demonstrated the potential therapeutic application of small molecule inhibitors in perturbation of the overactive TCA cycle, which promotes cancer progression [32,33]. 

Etiologically, the metabolic syndrome (MetS) has received increased attention in recent years for its active role in tumorigenesis, including HCC. Besides, obesity and diabetes have been identified as independent risk factors in HCC. These disorders show the presence of nonalcoholic fatty liver disease (NAFLD). This is characterized by excessive lipid accumulation in the hepatocytes due to enhanced levels of circulating free fatty acids (FFAs) in the blood plasma. Previous studies reported that NAFLD increased the risk of HCC in patients with viral and alcohol-related liver disease, which acted as the primary promoter of HCC [34]. Our previous study also showed that *Ras* oncogene-induced steatosis significantly enhanced the hepatic tumorigenesis in *Ras*-Tg [35]. Thus, the excessive hepatic lipid content and hepatocarcinogenesis are closely related.

In hepatoma cells, high intrinsic fat accumulation leads to a significant increase in the lipid drops (LDs) content in HCCs. Recent studies have demonstrated the beneficial functional role of LDs in cancerous cells. One such function is the prevention of endoplasmic reticulum stress due to misfolded proteins accumulation by promoting the temporary localization of misfolded proteins in LDs and eliminating misfolded proteins via proteasome regulation. Besides, LDs imparts HCC with remarkable drug resistance by lipophilic anti-tumor drugs’ sequestration and reactive oxygen species’ scavenging [31]. In *Ras*-Tg, the *Ras* oncogene-induced hepatic tumor is characterized by excessive LDs, which leads to the initiation and development of the liver tumor. Hence, the lipid level is also crucial for HCC development.

However, as compared to the primary HCC, advanced HCC with no massive LDs [36] demonstrates a negative role of heavy lipid in advanced and metastatic hepatoma cells. In *Ras*-Tg, the metastasis was rarely observed even in dying mice with massive hepatic tumors (data not shown). It indicated that *Ras*-Tg mouse is a significant animal model in the study of early-stage HCC but not advanced-stage HCC. LDs with a bigger dimension might burden and serve as cytomorphosis blocker for metastasis in hepatoma cells. Moreover, hepatic lipotoxicity has been demonstrated to be an inducer of hepatocellular death, oxidative and ER stress, insulin resistance, and so on [37]. The mechanism that balances the lipid utilization, LDs’ biogenesis, and export of excessive lipid for lipid cytotoxicity prevention must be explored. Apolipoprotein B (ApoB) is a primary lipoprotein for lipid export from hepatocytes and has been reported to be elevated in parallel with hepatic tumor sizes [38]. In *Ras*-Tg, ApoB was also significantly up-regulated in hepatic tumor cells (data not shown). Therefore, further investigations on the lipid metabolism balance system will provide crucial information for the development of precise HCC stage-specific therapies. 

## 5. Conclusions

In conclusion, the metabolomics-based expression-pattern analysis revealed dynamic changes of metabolites during *Hras12V*-induced hepatocarcinogenesis, which might elucidate the underlying mechanisms. By integrating metabolomics with the transcriptomics data, several important metabolic pathways were identified. The crucial role of the mitochondrial citrate-malate shuttle, which connected glycolysis to lipid biosynthesis in *Ras*-oncogene-induced hepatic tumorigenesis, was validated by in vivo assay of [U-^13^C]-labeled glucose. These findings will provide a platform for the development of precise therapeutic agents for HCC.

## Figures and Tables

**Figure 1 metabolites-10-00193-f001:**
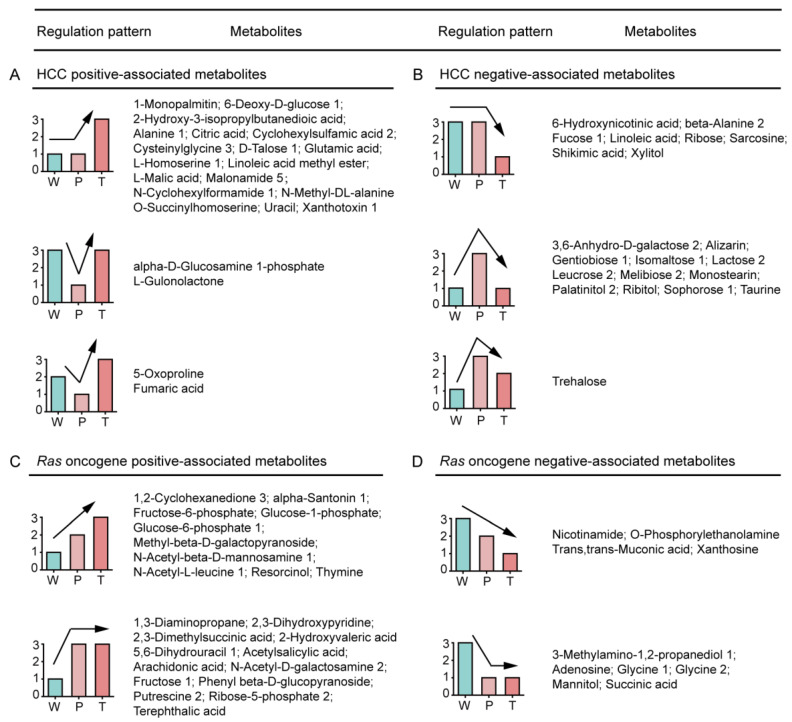
Expression-pattern analysis of W, P, and T metabolites in *Ras* oncogene-induced hepatocarcinogenesis. (**A**,**B**) The positively and negatively associated metabolites of HCC, respectively. (**C**,**D**) The positively and negatively associated metabolites of *Ras* oncogene, respectively. A detailed description is provided in Appendix A. W: normal liver tissue of wild-type mice; P: precancerous tissue of *Ras*-Tg; T: hepatocellular carcinoma tissue of *Ras*-Tg; *Ras*-Tg: *Hras12V* transgenic mice; HCC: hepatocellular carcinoma.

**Figure 2 metabolites-10-00193-f002:**
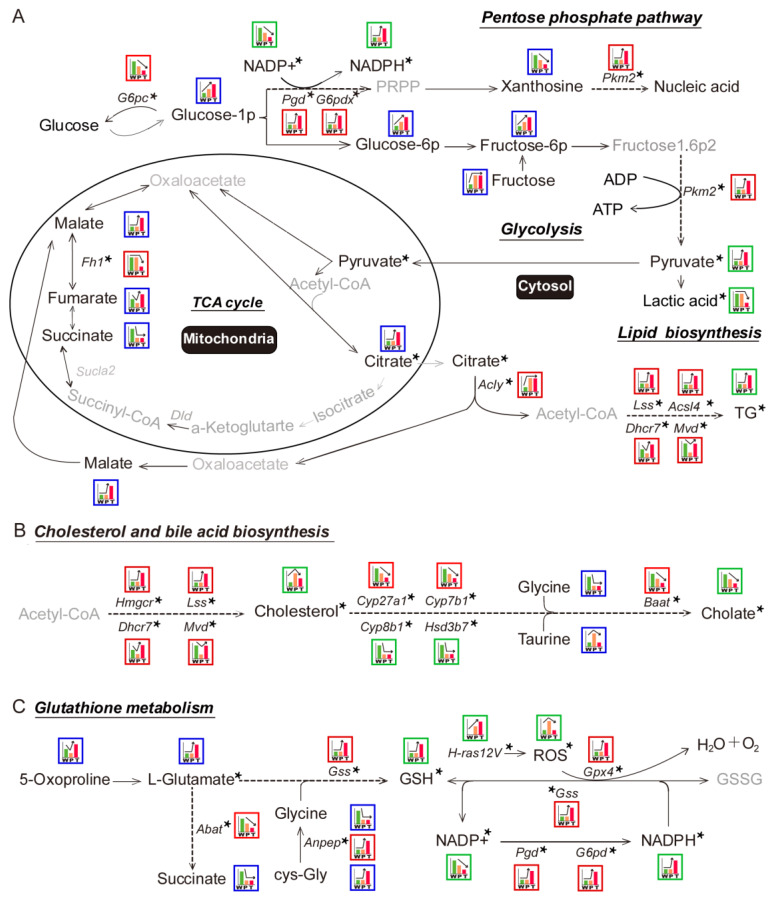
Schematic representation of the metabolic pathways in hepatocarcinogenesis. An exhaustive analysis of the integrated metabolomic and transcriptomic data identified multiple relevant metabolic pathways for further validation and investigation. (**A**) Metabolic pathways associated with glycolysis, tricarboxylic acid (TCA) cycle, pentose phosphate pathway (PPP), and lipid biosynthesis. (**B**) Metabolic pathways associated with cholesterol and bile acid synthesis. (**C**) Metabolic pathways associated with glutathione. The blue- and red-bordered boxes depict the expression-patterns of metabolomics and transcriptomics data, respectively. The green bordered boxes indicate the expression-patterns of metabolites that were detected directly. The relative expression levels of W (green histogram bar), P (orange histogram bar), and T (red histogram bar) are demonstrated in boxes; the different height of histogram bars indicate expression levels. The asterisk in the upper right corner of DEGs or DEMs indicates that these metabolites or enzymes were validated (detailed information is given in Figure 3 and Figure 4). Grey word represents the undetected metabolites in the present study. TG: triglyceride; NADPH: nicotinamide adenine dinucleotide phosphate; GSH: glutathione; ROS: reactive oxygen species. Other abbreviations are the same, as mentioned in Figure 1.

**Figure 3 metabolites-10-00193-f003:**
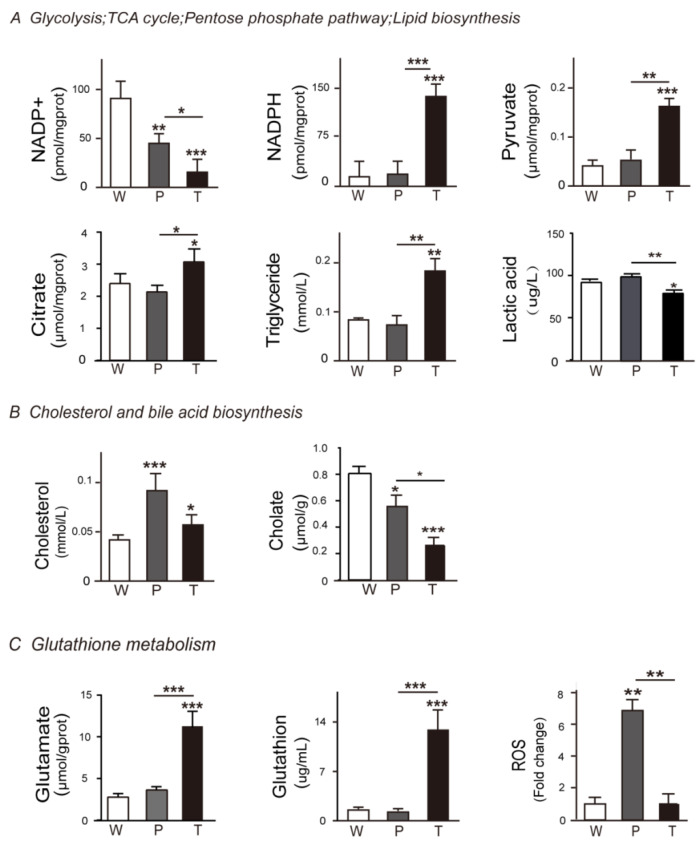
Validation of DEMs (differently expressed metabolites) in HCC, precancerous liver tissues, and normal liver tissues. (**A**) Glycolysis, TCA cycle, pentose phosphate pathway, and lipid biosynthesis correlated key DEMs were validated. (**B**) Cholesterol and bile acid biosynthesis correlated key DEMs were validated. (**C**) Glutathione metabolism correlated key DEMs were validated. The key metabolites involved in Figure 2 were validated using the corresponding methods described in the Appendix A. Abbreviations are the same, as mentioned in Figure 1.

**Figure 4 metabolites-10-00193-f004:**
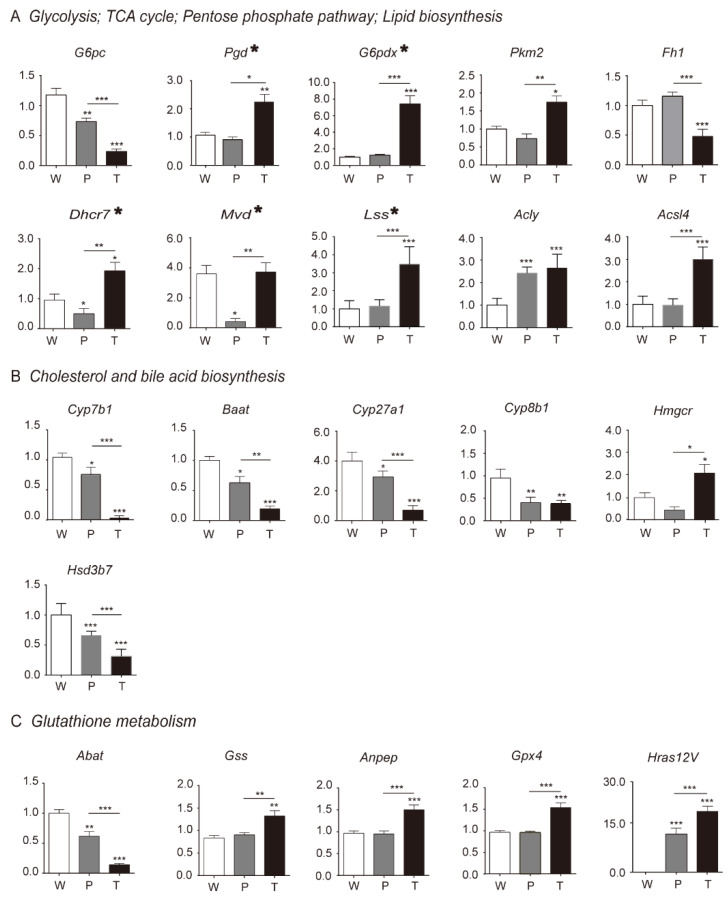
Validation of DEGs (differential expression of genes) in HCC, precancerous liver tissues, and normal liver tissues. The key DEGs involved in Figure 2 were validated using RT-qPCR assay at mRNA levels. (**A**) Glycolysis, TCA cycle, pentose phosphate pathway, and lipid biosynthesis correlated key DEGs were validated. (**B**) Cholesterol and bile acid biosynthesis correlated key DEGs were validated. (**C**) Glutathione metabolism correlated key DEGs were validated. The asterisk-marked genes in (**A**) indicate that they are also included in the pathways of (**B**) and (**C**). Abbreviations are the same, as mentioned in Figure 1.

**Figure 5 metabolites-10-00193-f005:**
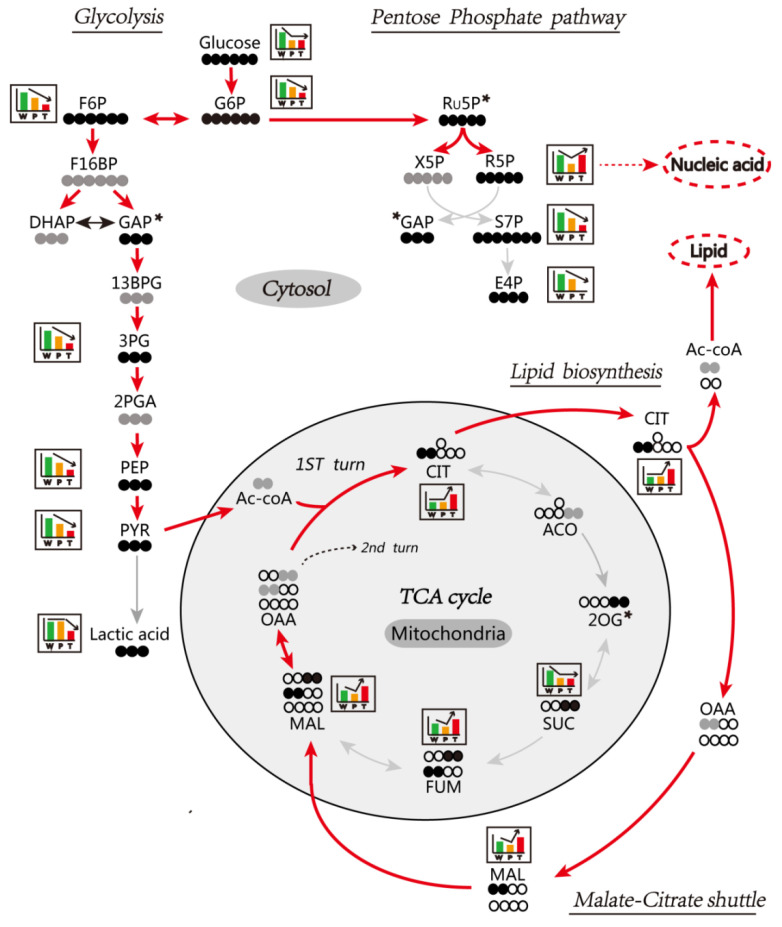
Schematic representation of [U-^13^C] glucose metabolic flux in the liver. The metabolic flux of [U-^13^C] glucose in the glycolysis, the pentose phosphate pathway, and the tricarboxylic acid (TCA) cycle is demonstrated. The solid and hollow black circles represent the ^13^C and ^12^C, respectively. The solid gray circles represent the ^13^C in undetected metabolites. The boxes with histograms represent the expression-patterns among W, P, and T (significant changes at least twice in pairwise comparison). The asterisks represent metabolites with no change or only a single significant change in a pairwise comparison. The histogram bar represents the relative expression levels of W (green histogram), P (orange histogram), and T (red histogram), and the bar’s height indicates a significant change in expression levels. G6P: glucose 6-phosphate; F6P: fructose 6-phosphate; F16BP: fructose-1,6-bisphosphate; DHAP: dihydroxyacetone phosphate; GAP: glyceraldehyde 3-phosphate; 13BPG: 1,3-bisphosphoglycerate; 3PG: 3-phosphoglycerate; 2PGA: 2-phosphoglycerate; PEP: phosphoenolpyruvate; PYR: pyruvate; Ac-CoA: acetyl CoA; CIT: citrate; ACO: aconitate; 2OG: 2-oxoglutarate; SUC: succinate; FUM: fumarate; MAL: malate; OAA: oxaloacetate; Ru5P: ribulose-5-phosphate; X5P: xylulose-5-phosphate; R5P: ribose-5-phosphate; S7P: sedoheptulose-7-phosphate; E4P: erythrose-4-phosphate. Other abbreviations are the same, as mentioned in Figure 1.

**Table 1 metabolites-10-00193-t001:** DEMs in paired comparisons between W, P, and T.

Metabolites	Retention Time (min)	Liver Tissues			Fold Change/*p*-Value		
		W Mean	P Mean	T Mean	W vs. P	P vs. T	W vs. T
**Glycolysis; TCA cycle; Pentose phosphate pathway; Lipid biosynthesis**							
Glucose-1p (glucose-1-phosphate)	16.7179	1.09 × 10^−1^	1.48 × 10^−1^	2.14 × 10^−1^	0.74/1.38 × 10^−2^	0.69/1.97 × 10^−2^	0.51/2.81 × 10^−3^
Xanthosine	24.2838	1.73 × 10^0^	1.10 × 10^0^	4.54 × 10^−1^	1.57/2.81 × 10^−3^	2.43/9.13 × 10^−4^	3.81/7.11 × 10^−5^
Glucose-6p (glucose-6-phosphate)	21.4070	1.69 × 10^−2^	4.56 × 10^−2^	1.68 × 10^−1^	0.37/6.57 × 10^−3^	0.27/3.77 × 10^−6^	0.10/1.50 × 10^−5^
Fructose-6p (fructose-6-phosphate)	21.2896	2.67 × 10^−2^	7.82 × 10^−2^	2.39 × 10^−1^	0.34/1.15 × 10^−3^	0.33/1.15 × 10^−2^	0.11/9.13 × 10^−4^
Fructose	17.8171	9.71 × 10^−1^	3.36 × 10^0^	3.30 × 10^0^	0.29/1.12 × 10^−2^	1.02/9.37 × 10^−1^	0.29/2.45 × 10^−2^
Citrate (citric acid)	17.2200	1.00 × 10^−2^	1.21 × 10^−2^	4.55 × 10^−2^	0.83/6.84 × 10^−1^	0.27/3.91 × 10^−2^	0.22/3.18 × 10^−2^
Succinate (succinic acid)	11.8994	1.40 × 10^−1^	6.65 × 10^−2^	6.21 × 10^−2^	2.11/4.43 × 10^−3^	1.07/7.65 × 10^−1^	2.26/2.41 × 10^−3^
Fumarate (fumaric acid)	12.2976	1.56 × 10^−1^	1.33 × 10^−1^	2.29 × 10^−1^	1.17/3.28 × 10^−2^	0.53/7.11 × 10^−5^	0.68/5.66 × 10^−4^
Malate (L-Malic acid)	13.8313	2.08 × 10^0^	1.81 × 10^0^	3.39 × 10^0^	1.15/9.40 × 10^−2^	0.53/2.68 × 10^−4^	0.61/1.45 × 10^−3^
**Cholesterol and bile acid biosynthesis**							
Glycine	11.8509	2.22 × 10^0^	1.47 × 10^0^	1.63 × 10^0^	1.51/3.34 × 10^−5^	0.90/5.04 × 10^−1^	1.37/2.26 × 10^−2^
Taurine	15.9588	4.25 × 10^−1^	7.66 × 10^−1^	3.44 × 10^−1^	0.55/4.97 × 10^−2^	2.23/3.18 × 10^−2^	1.24/4.32 × 10^−1^
**Glutathione metabolism**							
5-oxoproline	14.4595	1.49 × 10^−1^	1.10 × 10^−1^	3.16 × 10^−1^	1.35/3.48 × 10^−3^	0.34/1.60 × 10^−4^	0.47/1.97 × 10^−2^
L-glutamate (glutamic acid)	15.2758	6.10 × 10^−3^	5.40 × 10^−3^	1.55 × 10^−2^	1.13/4.75 × 10^−1^	0.35/1.47 × 10^−2^	0.39/2.02 × 10^−2^
Succinate (succinic acid)	11.8994	1.40 × 10^−1^	6.65 × 10^−2^	6.21 × 10^−2^	2.11/4.43 × 10^−3^	1.07/7.65 × 10^−1^	2.26/2.41 × 10^−3^
Glycine	11.8509	2.22 × 10^0^	1.47 × 10^0^	1.63 × 10^0^	1.51/3.34 × 10^−5^	0.90/5.04 × 10^−1^	1.37/2.26 × 10^−2^
cys-Gly (L-cysteinylglycine)	17.2563	5.49 × 10^−2^	4.74 × 10^−2^	8.88 × 10^−2^	1.16/5.06 × 10^−1^	0.53/7.09 × 10^−3^	0.62/1.79 × 10^−2^

Note: The DEMs (differently expressed metabolites) identified by PCA (principal component analysis) between wide-type liver tissues (W; *n* = 8), precancerous liver tissues (P; *n* = 8), and hepatocellular carcinomas (T; *n* = 8) in Figure 2 are shown. *p* < 0.05 means the significant difference. TCA: tricarboxylic acid.

**Table 2 metabolites-10-00193-t002:** DEGs in paired comparisons between W, P, and T.

mRNA Accession No.	Gene Symbol (Full Name)	Fragments Per Kilobase Million (FPKM)			Fold Change/*p*-Value		
		W	P	T	W vs. P	P vs. T	W vs. T
**Glycolysis; TCA cycle; Pentose phosphate pathway; Lipid biosynthesis**							
001199296	Acly (ATP-citrate lyase, transcript variant 1)	2.54 × 10^1^	4.57 × 10^1^	4.98 × 10^1^	0.56/2.45 × 10^−2^	1.09/9.89 × 10^−2^	0.51/1.09 × 10^−4^
010209	Fh1 (fumarate hydratase 1)	1.37 × 10^2^	1.24 × 10^2^	5.65 × 10^1^	1.10/4.58 × 10^−1^	0.45/9.75 × 10^−4^	2.42/6.23 × 10^−5^
008061	G6pc (glucose-6-phosphatase)	2.18 × 10^2^	9.61 × 10^1^	1.13 × 10^1^	2.27/5.01 × 10^−12^	0.12/3.62 × 10^−13^	19.33/1.98 × 10^−39^
008062	G6pdx (glucose-6-phosphate dehydrogenase X-linked)	1.42 × 10^0^	3.65 × 10^0^	1.69 × 10^1^	0.39/6.88 × 10^−1^	0.22/6.03 × 10^−4^	0.08/6.17 × 10^−5^
001081274	Pgd (6-phosphogluconate dehydrogenase)	7.74 × 10^0^	1.28 × 10^1^	1.00 × 10^1^	1.02/1.00 × 10^0^	3.43/8.58 × 10^−10^	0.30/9.60 × 10^−10^
011099	Pkm (pyruvate kinase, muscle)	5.32 × 10^0^	7.12 × 10^0^	2.58 × 10^1^	0.75/7.91 × 10^−1^	3.62/1.66 × 10^−4^	0.21/1.76 × 10^−5^
146006	Lss (lanosterol synthase)	2.29 × 10^1^	1.35 × 10^1^	5.84 × 10^1^	1.69/1.42 × 10^−1^	4.32/9.60 × 10^−11^	0.39/2.61 × 10^−7^
007856	Dhcr7 (7-dehydrocholesterol reductase)	4.91 × 10^1^	1.79 × 10^1^	6.65 × 10^1^	2.74/8.36 × 10^−5^	3.72/3.19 × 10^−11^	0.74/2.80 × 10^−3^
138656	Mvd (mevalonate kinase)	1.48 × 10^1^	3.57 × 10^0^	1.96 × 10^1^	4.14/1.16 × 10^−2^	5.50/9.23 × 10^−5^	0.75/1.25 × 10^−1^
50790	Acsl4 (acyl-CoA synthetase long-chain family member 4)	8.57 × 10^0^	1.22 × 10^1^	3.44 × 10^1^	0.70/2.01 × 10^−1^	0.36/5.76 × 10^−6^	0.25/9.19 × 10^−4^
**Cholesterol and bile acid biosynthesis**							
007519	Baat (bile acid-CoA: amino acid acyltransferase)	1.50 × 10^2^	1.09 × 10^2^	3.50 × 10^1^	1.37/1.30 × 10^−2^	0.32/3.45 × 10^−6^	4.28/6.06 × 10^−12^
024264	Cyp27a1 (sterol 26-hydroxylase)	1.94 × 10^2^	1.13 × 10^2^	2.69 × 10^1^	1.71/4.56 × 10^−6^	0.24/2.09 × 10^−9^	7.21/1.04 × 10^−23^
007825	Cyp7b1 (25-hydroxycholesterol 7-alpha-hydroxylase)	3.64 × 10^2^	1.91 × 10^2^	7.32 × 10^0^	1.90/1.38 × 10^−13^	0.04/4.09 × 10^−37^	49.71/7.72 × 10^−78^
010012	Cyp8b1 (7-alpha-hydroxycholest-4-en-3-one)	1.40 × 10^2^	1.06 × 10^1^	2.08 × 10^0^	13.17/4.69 × 10^−30^	0.20/1.13 × 10^−1^	67.17/1.61 × 10^−30^
133943	Hsd3b7 (3 beta-hydroxysteroid dehydrogenase type 7)	2.79 × 10^2^	7.90 × 10^1^	4.26 × 10^1^	3.53/1.30 × 10^−27^	0.54/6.85 × 10^−2^	5.47/2.05 × 10^−31^
008255	Hmgcr (3-hydroxy-3-methylglutaryl-coenzyme A reductase)	3.18 × 10^1^	2.31 × 10^1^	8.69 × 10^1^	1.38/2.88 × 10^−1^	3.76/6.22 × 10^−14^	0.37/8.95 × 10^−11^
**Glutathione metabolism**							
172961	Abat (4-aminobutyrate aminotransferase)	8.37 × 10^1^	4.87 × 10^1^	1.67 × 10^1^	1.72/2.31 × 10^−3^	0.34/4.02 × 10^−3^	5.02/1.61 × 10^−8^
008162	Gpx4 (phospholipid hydroperoxide glutathione)	1.83 × 10^2^	1.67 × 10^2^	3.12 × 10^2^	1.10/3.64 × 10^−1^	1.87/5.35 × 10^−22^	0.59/1.86 × 10^−18^
008486	Anpep(alanyl (membrane) aminopeptidase)	2.57 × 10^1^	2.93 × 10^1^	5.17 × 10^1^	0.88/6.89 × 10^−1^	1.76/3.11 × 10^−4^	0.50/4.85 × 10^−5^
008180	Gss (glutathione synthetase)	2.87 × 10^1^	3.21 × 10^1^	5.37 × 10^1^	0.90/7.04 × 10^−1^	1.68/4.86 × 10^−4^	0.53/8.80 × 10^−5^

Note: The mRNA levels of DEGs (differential expression of genes) associated with metabolism pathways in Figure 2 are shown. Wide-type liver tissues (W; *n* = 5), precancerous liver tissues (P; *n* = 5), and hepatocellular carcinomas (T; *n* = 5) were analyzed. *p*-values were calculated using the Student’s *t*-test. *p* < 0.05 and fold change less than 0.67 or large than 1.5 means the significant difference.

**Table 3 metabolites-10-00193-t003:** DEMs in paired comparisons between W, P, and T by the targeted metabolomics analysis incorporating [U-^13^C] glucose.

Metabolites	Liver Tissues			*p*-Value		
	W Mean	P Mean	T Mean	P vs. W	W vs. T	P vs. T
Phosphoenolpyruvic acid-M3	2.19 × 10^−1^	1.49 × 10^−1^	9.82 × 10^−2^	2.32 × 10^−4^	3.10 × 10^−7^	1.35 × 10^−3^
Lactic acid-M3	1.48 × 10^−1^	1.27 × 10^−1^	8.77 × 10^−2^	7.31 × 10^−2^	3.51 × 10^−5^	1.02 × 10^−3^
Citric acid-M2	1.47 × 10^−1^	1.44 × 10^−1^	1.54 × 10^−1^	2.71 × 10^−1^	1.06 × 10^−2^	5.26 × 10^−3^
Succinic acid-M2	1.68 × 10^−1^	1.14 × 10^−1^	1.21 × 10^−1^	2.45 × 10^−6^	6.11 × 10^−6^	7.27 × 10^−2^
Fumaric acid-M2	1.03 × 10^−1^	9.71 × 10^−2^	1.16 × 10^−1^	8.87 × 10^−3^	2.19 × 10^−5^	6.08 × 10^−8^
Malic acid-M2	1.30 × 10^−1^	1.23 × 10^−1^	1.37 × 10^−1^	1.06 × 10^−3^	2.98 × 10^−3^	7.25 × 10^−7^
Glyceraldehyde 3-phosphate-M3	7.55 × 10^−2^	6.31 × 10^−2^	5.79 × 10^−2^	1.77 × 10^−1^	6.51 × 10^−2^	5.70 × 10^−1^
Erythrose 4-phosphate -M4	7.02 × 10^−3^	1.64 × 10^−3^	0.00 × 10^0^	1.07 × 10^−3^	8.30 × 10^−5^	2.00 × 10^−5^
Ribulose-5-phosphate -M5	2.78 × 10^−3^	2.00 × 10^−3^	1.83 × 10^−3^	4.34 × 10^−1^	2.52 × 10^−1^	8.79 × 10^−1^
Glucose 6-phosphate -M6	1.99 × 10^−1^	1.22 × 10^−1^	4.87 × 10^−2^	5.31 × 10^−3^	5.67 × 10^−7^	1.21 × 10^−3^
Sedoheptulose 7-phosphate -M7	7.26 × 10^−3^	4.71 × 10^−3^	1.20 × 10^−3^	2.29 × 10^−2^	2.42 × 10^−6^	8.75 × 10^−5^
Glucose-M6	2.65 × 10^−1^	1.63 × 10^−1^	1.36 × 10^−1^	2.60 × 10^−5^	3.79 × 10^−6^	2.04 × 10^−1^
Ribose 5-phosphate -M5	5.05 × 10^−3^	2.75 × 10^−3^	5.11 × 10^−3^	7.39 × 10^−3^	9.44 × 10^−1^	2.17 × 10^−2^
Fructose 6-phosphate -M6	1.70 × 10^−1^	1.01 × 10^−1^	3.78 × 10^−2^	1.11 × 10^−4^	2.66 × 10^−11^	1.85 × 10^−4^

Note: The DEMs identified by in vivo targeted metabolomics incorporating [U-^13^C] glucose analysis between wide-type liver tissues (W; *n* = 8), precancerous liver tissues (P; *n* = 8), and hepatic carcinomas (T; *n* = 8) in Figure 3 are shown. The mean values of liver tissues were the intensities of the labeled metabolites presented as the ratio of the peak area of the labeled metabolite to the total peak area pool of that metabolite ([labeled]/[total]). *p* < 0.05 means the significant difference. M plus Arabic numerals indicates the number of ^13^C-labeled carbon.

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
