# Peer review of "Role of the Mitochondrial Citrate-malate Shuttle in Hras12V-Induced Hepatocarcinogenesis: A Metabolomics-Based Analysis"

_metabolites, 2020, doi:10.3390/metabo10050193_

Round 1

Reviewer 1 Report

The revised version of the manuscript is adequate.

Reviewer 2 Report

The authors addressed all my concerns in the revised manuscript. The manuscript is much improved now, and I have no further comments.

Reviewer 3 Report

The authors have thoroughly considered and answered our suggestions in detail and performed additional experiments to conclusively demonstrate the changes in enzyme levels by Western blotting. Therefore, we now recommend this manuscript for publication.

This manuscript is a resubmission of an earlier submission. The following is a list of the peer review reports and author responses from that submission.

Round 1

Reviewer 1 Report

The authors investigated that the roles of metabolic pathways on hepatocarcinogensis using Ras-Tg mice. They concluded that mitochondrial citrate-malate shuttle was a crucial pathway during hepatocarcinogenesis in Hras12V mice.

  1. The results of histology should be indicated appropriately. How was the incidence and multiplicities for precancerous or HCC? The histological criteria was not described in Materials and Methods.
  2. The authors stated that only pathologically diagnosed tissues were used for following analyses. However the histology was assessed by the remaining liver tissue. How were normal, precancerous and tumor separated completely?

Reviewer 2 Report

In this manuscript, the authors bring evidence for the crucial role of the mitochondrial citrate-malate 
shuttle in Hras12V induced hepatocarcinogenesis. The topic is new and the study is very interesting and properly carried out. 
However, below are some suggestions to further improve the paper.

Figure 2: Replace mitochondrial ACLY with Citrate Synthase (CS)

               Please correct this mistake everywhere in the text too.

Table 2: Since the authors found increased citrate levels and increased          ACLY levels in P and/or T as compared to W group, I think they need to quantify SLC25A1 mRNA too and describe the obtained results in the text as this mitochondrial carrier is the specific transporter of citrate/malate.

Discussion - Pag 14 - Lines 385-398: Discussing the crucial role of TCA cycle in tumorigenesis, the authors must consider the recent and interesting review about the TCA cycle rewiring in HCC (Cancers 2020, 12, 68; doi:10.3390/cancers12010068)

Line 386 : replace “pruvat” with “pyruvate” 

Reviewer 3 Report

  1. In the introduction, some claims made by the authors lack sufficient clarity. For example: “molecular heterogeneity in the cancer cells” (line 40) – does this refer to cellular heterogeneity within a tumor, e.g. different cells/cell types behaving differently? Or just that there are many heterogeneous molecules in the cell?
  2. Some claims are too categorical and dismissive of large areas of cancer metabolism field, e.g. “Previous studies have primarily targeted the role of the Warburg effect in cancer; however, the novel and other relevant biochemical changes in cancer were never unveiled.” (lines 42-43). In contrast, many seminal studies on pathways other than glycolysis have been performed - for example, on the role of glutaminolysis, serine biosynthesis/one-carbon pathway and 2-hydroxyglutarate production in cancer, among many others.
  3. Line 64: “In murine HCC, Ras mutation was reported in 70% of cases” Is this DEN-induced carcinogenesis? A reference should be provided here.
  4. The authors should refrain from overinterpreting their data and drawing far-reaching conclusions. For example, lines 224-231: the authors state that HCC has increased glycolysis, citrate-malate shuttle and lipid biosynthesis purely on the basis of some of the metabolites and transcripts for metabolic enzymes being elevated. These conclusions are drawn even before the authors present the [U-13C] glucose studies. But an increase in any given metabolite could also arise from their reduced utilization – e.g. increased citrate and high NADPH/NADP+ could arise from utilization of these metabolites in fatty acid biosynthesis, and elevated triglycerides can reflect increased uptake instead. How can authors rule these out? Furthermore, drawing conclusions from transcript levels of select metabolic enzymes is not sufficient evidence that these enzymes are actually elevated in these samples. The authors should perform Western blots for enzyme levels, because transcript levels often do not correlate well with actual protein levels of metabolic enzymes. Similarly, the title of the paper is overstating the findings and needs to be revised. To determine that citrate-malate shuttle plays a crucial role, genetic and/or inhibitor studies are needed (which are understandably beyond the scope of this work).
  5. How was the [U-13C] Glucose metabolic flux analysis performed? What do the numbers in Table 5 represent – are these relative abundances of 13C-labeled metabolites normalized to tissue weight? There are no units provided. It would have been more useful to represent them as fractions of the total pool of indicated metabolites, e.g. M+3 citrate fraction of total citrate pool. This way, an idea of turnover of these pools can be better understood – e.g. from Table 3 it is clear that citrate is up in P and T relative to normal tissue, but from Table 5, M+3 citrate levels don’t differ much. This looks like the conversion of glucose carbons into citrate is actually faster in normal tissue than in P and T? Overall, representing the data as [labeled]/[total] may be a more useful metric rather than looking at absolute values of how much 13C carbon has reached the citrate pool in 45 minutes post-infusion.
  6. Line 234: “Hydrogen (H) is a crucial component in the synthesis of biomolecules” – it would be more correct to say “hydride anion” as it is the extra electron it carries that is actually used in the synthesis.
  7. One striking finding is the abundance of ROS in precancerous lesions which is reverted back to the level of that tin the normal tissue in tumor and is associated with a dramatic increased in glutathione and NADPH/NADP+ ratio. This looks like ROS-associated stress is a key component of the selective pressure exerted upon precancerous lesions. It also looks like in order to complete the transformation process from a precancerous lesion to a carcinoma, cells must mount robust antioxidant defenses first, which is worth further investigating on a genomic/transcript/proteomic level.
  8. “Supplication” should be “supply” (word choice).

Taken together, the manuscript and its conclusions should be considerably revised, flux analysis data should be better presented and Western blots for at least some metabolic enzymes that authors find elevated need to be performed, before this work can be considered ready for publication.